# Neurotherapeutic Effect of *Inula britannica* var. Chinensis against H_2_O_2_-Induced Oxidative Stress and Mitochondrial Dysfunction in Cortical Neurons

**DOI:** 10.3390/antiox10030375

**Published:** 2021-03-03

**Authors:** Jin Young Hong, Hyunseong Kim, Junseon Lee, Wan-Jin Jeon, Seung Ho Baek, In-Hyuk Ha

**Affiliations:** 1Jaseng Spine and Joint Research Institute, Jaseng Medical Foundation, Seoul 135-896, Korea; vrt3757@gmail.com (J.Y.H.); biology4005@gmail.com (H.K.); excikind@gmail.com (J.L.); poghkl@gmail.com (W.-J.J.); 2College of Korean Medicine, Dongguk University, 32 Dongguk-ro, Ilsandong-gu, Goyang-si 10326, Korea; Baekone99@gmail.com

**Keywords:** *Inula britannica* var. chinensis, hydrogen peroxide, brain-derived neurotrophic factor, nerve growth factor, neuroprotection, synaptophysin, nerve regeneration

## Abstract

*Inula britannica* var. chinensis (IBC) has been used as a traditional medicinal herb to treat inflammatory diseases. Although its anti-inflammatory and anti-oxidative effects have been reported, whether IBC exerts neuroprotective effects and the related mechanisms in cortical neurons remain unknown. In this study, we investigated the effects of different concentrations of IBC extract (5, 10, and 20 µg/mL) on cortical neurons using a hydrogen peroxide (H_2_O_2_)-induced injury model. Our results demonstrate that IBC can effectively enhance neuronal viability under in vitro-modeled reaction oxygen species (ROS)-generating conditions by inhibiting mitochondrial ROS production and increasing adenosine triphosphate level in H_2_O_2_-treated neurons. Additionally, we confirmed that neuronal death was attenuated by improving the mitochondrial membrane potential status and regulating the expression of cytochrome c, a protein related to cell death. Furthermore, IBC increased the expression of brain-derived neurotrophic factor and nerve growth factor. Furthermore, IBC inhibited the loss and induced the production of synaptophysin, a major synaptic vesicle protein. This study is the first to demonstrate that IBC exerts its neuroprotective effect by reducing mitochondria-associated oxidative stress and improving mitochondrial dysfunction.

## 1. Introduction

Adenosine triphosphate (ATP), which carries energy for intracellular metabolism, is synthesized in the mitochondria and is mostly used in the cytoplasm [1]. The mitochondria also play important physiological roles in maintaining cellular homeostasis such as cell growth, cell cycle regulation, production of reactive oxygen species (ROS), and cell death [2,3]. In neurons, the mitochondria are found in the cytoplasm and along the axon [4]. When neurons are injured, most of their mitochondria also sustain an injury, and ATP production is either reduced or stopped. Additionally, when morphological and functional mitochondrial changes occur, most of the oxygen is used to produce ROS. The excessive ROS produced induces DNA damage in the cells and creates oxidative stress that induces apoptosis and tissue injury [1,5,6]. Previous studies have reported that oxidative stress plays a key role in the pathophysiological mechanism of cardiovascular, degenerative, and neurological diseases [7,8]. Thus, reducing oxidative stress by regulating mitochondrial ROS production is critical. In this regard, recent studies have shown that excessive ROS production and mitochondrial morphological and functional changes are major factors in creating an oxidative environment [9,10]. A previous study identified an association between neurodegenerative diseases, such as Alzheimer’s, Parkinson’s, and Huntington’s diseases, and impaired mitochondrial homeostasis [11,12]. Another study indicated that mitochondria regulate programmed neuronal death. Furthermore, mitochondria regulate the synthesis of neurotransmitters, which are closely associated with neurological diseases, maintenance and formation of synapses, neurogenesis and development, axonal transport, and neuroplasticity [13,14].

Active research is underway to develop natural product-based therapies that utilize mitochondrial functions for the treatment of neurological diseases. Such products are of little concern for adverse effects and have already been empirically verified for safety and efficacy. Natural product-based treatments have already been used in primary healthcare, demonstrating excellent safety, a short development period, and a high probability of success [15]. Interest in this treatment type has increased due to the increase in studies related to its efficacy as primary therapy in a wide range of diseases, including critical illnesses. Therefore, we aimed to examine a natural product with either neuroprotective or neuroregenerative effects that pose little risk of adverse effects. *Inula britannica* var. chinensis (IBC) flower extract have been used as the main herbal ingredient used in jaseng soongiwon which is an herbal prescription for treatment of stroke.

Our findings indicate that IBC flower extract exerts its neuroprotective effect by restoring mitochondrial function and reducing oxidative stress. This is the first study to confirm the neuroprotective and neuroregenerative effects of IBC extract through the restoration of mitochondrial function in cortical neurons injured by exposure to H_2_O_2_.

## 2. Materials and Methods

### 2.1. In Vitro Culture of Cortical Neurons

All animals used for this study were maintained and handled in accordance with Jaseng Animal Care and Use Committee guidelines (JSR-2020-03-004). Primary cortical neurons were prepared from Sprague–Dawley rat embryos (embryonic day 17). Briefly, the isolated cortices were placed in Hank’s balanced salt solution (HBSS; Gibco BRL, Grand Island, NY, USA), and the meninges were manually removed from the cerebral hemispheres. The tissues were rinsed twice in HBSS medium and then digested with 2 mL of 2.5 mg/mL papain solution (Sigma-Aldrich, St. Louis, MO, USA) in HBSS for 15 min at 37 °C. After digestion, the supernatant was discarded, and the tissues were rinsed twice in 2 mL HBSS and centrifuged at 1500 rpm for 3 min to obtain the cell pellet. The cells were resuspended in 1 mL cortical neuron culture media containing Neurobasal medium (Gibco) supplemented with B27 (Gibco), Gluta-MAX (Gibco), and 1% penicillin/ streptomycin (Gibco). The single cells were then seeded onto 12-mm circular coverslips (Paul Marienfeld GmbH & Co., Lauda-Königshofen, Germany) for immune staining, 6-well plates for fluorescence-activated cell sorting (FACS) analysis, and 96-well plates for the cell viability assay. The plates were prepared by coating the wells with 20 mg/mL poly-D-lysine (Gibco) overnight and 10 mg/mL laminin (Gibco) for 2 h at 4 °C.

### 2.2. Preparation of I. britannica var. Chinensis (IBC) Extract

The IBC extract was prepared according to a previously described method [16]: 300 g of IBC was heated to 105 °C in 3 L of distilled water for 3 h. After cooling at −20 °C for 30 min, the mixture was filtered once through a filter paper (HA-030, Hyundai Micro, Seoul, Korea) and then lyophilized in a freeze dryer (Ilshin BioBase, Gyeonggi-do, Korea) to obtain IBC dry extract. The extract yield was calculated and then re-dissolved in phosphate-buffered saline (PBS) to the desired concentration. It was stored at −20 °C until use.

### 2.3. Hydrogen Peroxide (H_2_O_2_)-Induced Neuronal Injury and IBC Treatments

The rat primary cortical neurons were plated at different densities on poly-D-lysine/laminin plates: 2 × 10^4^ cells/90 µL in 96-well plates for neuronal viability assessment; 4 × 10^5^ cells/450 µL in 24-well plates for immunocytochemistry (ICC); 2 × 10^6^ cells/1.8 mL in 6-well plates for flow cytometry; 4 × 10^6^ cells/2.7 mL in 60-mm^2^ dishes for real-time PCR. After allowing 2 h for attachment and stabilization, H_2_O_2_ was added to the cortical neurons to reach a final concentration of 500 µM, and the cells were further incubated for 30 min. After this, the H_2_O_2_-containing medium was discarded and replaced with a new cortical culture medium containing various concentrations of IBC extract (5, 10, 20 µg/mL). The plates were then incubated in 5% CO_2_ at 37 °C for 24 h (Figure 1A shows the schematic diagram illustrating the experimental design). In addition, to assess the synaptic density and longest neurite length by use of a synaptic marker for ICC, the cells were prepared in 24-well plates at a density of 4 × 10^5^ cells/450 µL and incubated for 7 days in vitro (DIV). The 7 DIV cortical neurons were treated and incubated for 24 h as described in Figure 1A.

### 2.4. Neuronal Viability Assays

The cells were assessed after 24 h of the IBC extract treatment. Cellular viability was analyzed using the Cell Counting Kit-8 (CCK-8; Dojindo, Kumamoto, Japan). CCK-8 solution (10 µL) was added to each well and incubated for 4 h at 37 °C. Absorbance at 450 nm was then measured using a microplate reader (Epoch, BioTek, Winooski, VT, USA). Cell viability was expressed as a percentage of the non-treated cells, which were defined as 100% viable. In addition, cell viability was also determined by the LIVE/DEAD Cell Imaging Kit (Thermo Fisher Scientific, Waltham, Massachusetts, USA), in accordance with the manufacturer’s instructions. Briefly, the staining solution consists of two probes that assess recognized parameters of cytotoxicity and cell viability, a Calcein AM dye that turns fluorescent green in live cells, and BOBO-3 Iodide that stains dead cells with red fluorescence. To stain the cells, the culture medium was discarded, and each sample was incubated in 100 µL staining solution for 15 min at 20–25 °C. Ten images were randomly acquired from each group at 10× magnification via confocal microscopy (Eclipse C2 Plus, Nikon, Minato City, Tokyo, Japan). The live and dead cells were manually counted on the images using ImageJ software (1.37 v, National Institutes of Health, Bethesda, MD, USA).

### 2.5. Mitochondrial Staining

MitoSOX-based assays are commonly used to quantify cellular ROS, particularly mitochondrial superoxide, in live cells. Primary cortical neurons were prepared on glass coverslips in 24-well plates and treated as described in Figure 1A. After 24 h of incubation, the cells were washed twice with HBSS/Ca++/Mg++ (Gibco) and treated with 5 µM MitoSOX Red mitochondrial superoxide indicator (Thermo Fisher Scientific) for 10 min at 37 °C in the dark. After washing with HBSS medium, the stained samples were mounted onto glass slides using Dako Mounting Medium (DAKO, Carpinteria, CA, USA). Images were captured via confocal microscopy (Eclipse C2 Plus, Nikon). Additionally, we confirmed the mitochondrial morphological changes using MitoTracker Red CMXRos (Molecular Probes Inc., Eugene, OR, USA) and MitoTracker Green FM (Thermo Fisher Scientific). Briefly, the plated and treated neurons were stained with a pre-warmed staining solution containing 200 nM MitoTracker Red or Green and incubated for 30 min in an incubator with 5% CO_2_. After staining, the staining solution was discarded, and the neurons were fixed with 4% paraformaldehyde (PFA) for additional labeling with Tuj 1 (neuron-specific class III beta-tubulin, 1:2000; R&D systems, McKinley Place NE, USA), in accordance with previously published protocols [17]. The mitochondrial morphology in neurons was observed via confocal microscopy (Eclipse C2 Plus, Nikon).

### 2.6. Mitochondrial ATP Assay

The mitochondria synthesize ATP for cell respiration and metabolic homeostasis. ATP level was assessed using the CellTiter-Glo Luminescent Cell Viability Assay (Promega, Madison, WA, USA), in accordance with the manufacturer’s instructions. All assays were performed in opaque-walled 96-well plates with the cells in a culture medium. Briefly, the cells were prepared in opaque-walled 96-well plates at a density of 2 × 10^4^ cells/well and were treated and incubated for 24 h as described in Figure 1A. The CellTiter-Glo reagent was added to the wells at a 1:1 ratio (*v/v*). The contents were mixed for 2 min in an orbital shaker to induce cell lysis. The plates were incubated at room temperature for 10 min to stabilize the luminescent signal and were then examined on a GloMax Navigator Microplate Luminometer (Promega). The ATP standard curve was prepared using serial ten-fold dilutions of an ATP solution in a culture medium to which CellTiter-Glo reagent was added to each well at a 1:1 (*v/v*) ratio. The luminescence values were determined using the standard curve.

### 2.7. Mitochondrial Membrane Potential (ΔΨm) Assays

Mitochondrial membrane potential (ΔΨm, MMP) is an important mitochondrial function parameter and serves as a cellular apoptosis inducer. JC-1, a sensitive mitochondrial fluorescent dye, was used to measure the MMP. In healthy cells with high mitochondrial membrane potential, JC-1 aggregates in the mitochondria and emits red fluorescence (λem  =  590 nm). Unhealthy or apoptotic cells have low mitochondrial membrane potential; thus, JC-1 monomers show intense green fluorescence (λem  =  527 nm). MMP was analyzed using the MitoProbe™ JC-1 Assay Kit (Thermo Fisher Scientific), in accordance with the manufacturer’s protocol. Briefly, primary cortical neurons were seeded onto 24 well plates. When cells adhered to the bottom of the culture plate, they were treated and incubated for 24 h, as described in Figure 1A. The cells were then stained with 2 µg/mL JC-1 at 37 °C and incubated for 30 min in the dark. They were then washed twice with PBS, and JC-1 fluorescence was immediately observed via confocal microscopy (Eclipse C2 Plus, Nikon). Additionally, we assessed the MMP using tetramethylrhodamine (TMRM; Thermo Fisher Scientific), a methyl ester fluorescence method. TMRM is a cell-permeant dye that accumulates in active mitochondria with intact membrane potentials. Primary cortical neurons were stained with 10 nM TMRM in HBSS for 30 min at 37 °C in the dark. After washing twice with PBS, images were acquired using confocal microscopy (Eclipse C2 Plus, Nikon), and the data were quantified and processed using ImageJ software (National Institutes of Health).

### 2.8. Immunocytochemistry

Immunocytochemistry was performed in the H_2_O_2_-activated cortical neurons. After treating with IBC extract at the different concentrations, the samples were fixed with 4% PFA for 30 min and rinsed three times for 5 min each with PBS (Gibco). The cells were incubated with 0.2% Triton X-100 in PBS for 5 min, rinsed twice with PBS for 5 min, and blocked with 2% normal goat serum (NGS) in PBS for 1 h. The following primary antibodies were used: 8-hydroxy-2’-deoxyguanosine (8-OHdG; 1:200; Santa Cruz Biotechnology, Santa Cruz, CA, USA), cytochrome c (CycC; 1:100; BD Biosciences, San Jose, CA, USA), microtubule-associated protein 1 light chain 3β (LC3B; 1:200; Novus, Littleton, CO, USA), SQSTM1/p62 (p62; 1:100; Abcam plc, Cambridge, MA, USA), nuclear respiratory factor 2 (Nrf2; 1:200; Abcam plc), brain-derived neurotrophic factor (BDNF; 1:200; Abcam plc), nerve growth factor (NGF; 1:100; Abcam plc), synaptophysin (Syn; 1:500; Sigma-Aldrich), and Tuj 1 (1:2000; R&D system). The primary antibodies were diluted in 2% NGS and incubated overnight at 4 °C. After washing with PBS three times for 5 min each, the samples were then incubated for 2 h with fluorescent secondary antibodies (FITC-conjugated goat anti-rabbit IgG or FITC-conjugated goat anti-mouse IgG, Jackson ImmunoResearch Laboratories) diluted at 1:300 in 2% NGS. After 2 h of incubation at room temperature, the cells were washed three times for 5 min with PBS, mounted with fluorescence mounting medium (Dako Cytomation), and images were acquired by confocal microscopy (Eclipse C2 Plus, Nikon). Ten representative images were captured at 100× or 400× magnification with the same acquisition settings via confocal microscopy. The average fluorescence intensity was measured using ImageJ software (National Institutes of Health) and compared quantitatively. Synaptic density was quantified by the number of syn-positive pixels, using the ImageJ software. Then, the number of syn-positive pixels was divided by the total number of pixels to calculate the mean percent syn labeled pixels.

### 2.9. Genomic, Mitochondrial DNA Preparation, and Dot Blot Assay

We performed a DNA dot blot assay to investigate the effect of IBC on oxidative DNA damage in nuclear and mitochondrial DNA (nDNA and mtDNA). 8-OHdG, an oxidized DNA nucleoside, is the most frequently detected nucleoside in nDNA and mtDNA. In brief, nDNA was extracted and isolated from the primary cortical neurons in each group using a DNeasy Blood and Tissue Kit (QIAGEN, Hilden, Germany) to quantify 8-OHdG (n = 3 per group). mtDNA was also prepared using a mitochondrial isolation kit (Abcam plc). Purified nDNA and mtDNA samples were spotted on a nitrocellulose membrane (0.2 µm pore size) and hybridized to the membrane by baking it at 80 °C for 2 h. The membrane was then blocked with 5% skim milk and incubated with mouse anti-dsDNA (1:2000, Abcam plc) and 8-OHdG (1:200, Santa Cruz Biotechnology) at room temperature (RT) for 1 h. Samples were then incubated with an anti-mouse or anti-rabbit horseradish-peroxidase-conjugated antibody (1:1000, Jackson ImmunoResearch Laboratories) for 1 h at RT. The antibody binding was visualized based on enhanced chemiluminescence.

### 2.10. Real-Time PCR

We analyzed changes in the expression levels of mitochondria and growth-related genes in each group using real-time PCR. Total RNA was isolated using TRIzol reagent (Ambion, Thermo Fisher Scientific). cDNA was synthesized using random hexamer primers and Accupower RT PreMix (Bioneer, Daejeon, Korea). All primer pairs were designed using the UCSC Genome Bioinformatics and the NCBI databases, and their sequences are listed in Table 1. Real-time PCR was performed using the iQ SYBR Green Supermix (Bio-Rad, Hercules, CA, USA) on a CFX Connect Real-Time PCR Detection System (Bio-Rad). All real-time PCR reactions were performed in triplicate or more. The expression of the target genes was normalized to GAPDH and expressed as the fold change relative to the control group.

### 2.11. PCR-Based Analysis of mtDNA/nDNA Ratio

mtDNA concentration was analyzed using the quantitative ratio of the target mitochondrial gene and reference nuclear gene (mtDNA/nDNA), by quantitative real-time PCR as described previously [18]. The genomic DNA was extracted from the cortical neurons using the DNeasy Blood and Tissue Kit (QIAGEN, Hilden, Germany) according to the manufacturer’s instructions. The mtDNA/nDNA ratio was measured by quantifying the number of mtDNA molecules to nDNA molecules with the classical ΔΔCt method used for qPCR analysis, and the primers used for the reaction were: mitochondrial- NADH-ubiquinone oxidoreductase chain 1 (mt-ND1) forward: 5′-CCGTCCTCCTAATAAGCGGC-3′ and mt-ND1 reverse: 5′-TATGGCTATTGGTCAGGCGG-3′; and GADPH forward: 5′-CCCCCAATGTATCCGTTGTG-3′ and GAPDH reverse: 5′-TAGCCCAGGATGCCCTTTAGT-3′.

### 2.12. Western Blotting

Total proteins were extracted in each group using RIPA buffer (CellNest, Minato, Tokyo, Japan) with Protease Inhibitor Cocktail Set III (1:1000, Millipore, Billerica, MA, USA). Protein concentration was measured using the Pierce BCA Protein Assay Kit (Thermo Fisher Scientific), in accordance with the manufacturer’s protocol. Protein samples were separated by SDS-PAGE, transferred to a polyvinylidene difluoride (PVDF) membrane, blocked with 5% Difco^TM^ skim milk (BD Biosciences) in 1X Tris-buffered saline (TBS, Bio-rad) with 0.1% Tween 20 (Sigma), and probed using various antibodies. The Western blots were visualized using ECL (Bio-Rad) and exposed to the Amersham Imager 600 (GE Healthcare Life Sciences, Uppsala, Sweden). The equivalence of protein loading was verified by probing for actin. The antibodies used were as follows: rabbit anti-PTEN-induced putative kinase 1 (PINK1, 1:500, Novus Biologicals, CO, USA); mouse anti-parkin (1:1000, Cell Signaling); mouse anti-β-actin (1:1000, Santa Cruz); horseradish peroxidase-conjugated anti-rabbit or -mouse antibodies (1:2500, Abcam plc).

### 2.13. Flow Cytometry

A flow cytometric assay was used to investigate the mitochondrial levels of superoxide and apoptosis. Apoptotic cell death was detected using an Annexin V-PE/PI apoptosis detection kit (Abcam plc) as previously described. Briefly, the cells were collected and incubated with 5 µL of Annexin V-PE and 5 µL of propidium iodide (PI) in 500 µL of 1× binding buffer and directly analyzed by FACS (Accuri C6 Plus Flow Cytometer, BD Bioscience). MitoSOX red mitochondrial superoxide indicator (Thermo Fisher Scientific) was used to assess mitochondrial oxidation. The cells were collected after treatment as described in Figure 1A, suspended in 100 µL of MitoSOX solution, and directly analyzed via flow cytometry. In addition, changes in Mitotracker, TMRM, BDNF, and NGF were analyzed in cortical neurons following IBC treatment with or without H_2_O_2_ exposure. Additionally, cellular ROS level was measured in cell lysate using 2′,7′–dichlorofluorescin diacetate (DCFDA; Sigma), a cell-permeant reagent that converts DCFDA which is oxidized to a fluorescence dye 2′,7′–dichlorofluorescein (DCF) to measure ROS activity within the cell [19]. The mean positive cell values, as determined via flow cytometry, were expressed as percent relative to the control group.

### 2.14. Statistical Analysis

The results were examined as means ± standard error of the mean using Prism software (GraphPad, San Diego, CA, USA). Multiple comparisons among the five groups were analyzed via one-way analysis of variance (ANOVA) with Tukey’s post hoc test. Differences were considered statistically significant if # *p* < 0.001 vs. the blank group and * *p* < 0.05, ** *p* < 0.01, or *** *p* < 0.001 vs. the H_2_O_2_ group.

## 3. Results

### 3.1. IBC Extract Potentially Protects against H_2_O_2_-Induced Oxidative Stress Injury in Rat Primary Cortical Neurons

A schematic illustration shows the experimental design of this study (Figure 1A). Cortical neurons were treated with IBC (0, 5, 10, 20 µg/mL) 24 h after being subjected to 500 µM H_2_O_2_ for 30 min to examine the neuroprotective and antioxidative effect through the improvement of mitochondrial biogenesis by immunological and molecular-biological analysis. When setting the optimal H_2_O_2_ concentration and incubation period to induce oxidative stress, neuronal viability was confirmed by ICC for concentrations of 100 and 500 μM H_2_O_2_ with an incubation period of 30 min and 1 h (Appendix A). The incubation of neurons with 100 μM H_2_O_2_ for 30 min induced a mild neuronal injury. However, the more severe neuronal injury was induced by exposure to 500 μM H_2_O_2_ for 30 min. Substantial neuronal injury was detected, with very low viability, at 1 h after initiating treatment with both 100 and 500 μM H_2_O_2_. We finally chose the concentration and incubation period of 500 μM and 30 min, respectively, as suggested by a previous report [20], to induce oxidative stress in cortical neurons.

Primary cortical neurons were first treated with various IBC extract concentrations for 24 h without H_2_O_2_ treatment to assess whether IBC extract exerted any cytotoxic effect. The CCK-8 assays confirmed that IBC extract concentrations of up to 50 µg/mL were safe to use on primary cortical neurons in vitro (Figure 1B). In subsequent experiments, the IBC extract at the tested concentrations (5–50 µg/mL) exerted a significant protective effect against H_2_O_2_-induced neuronal toxicity (Figure 1C). The LIVE/DEAD cell assay performed under the same culture conditions revealed a trend similar to that observed in the CCK-8 assay (Figure 1D,E). The number of green-stained viable cortical neurons was significantly lower in the H_2_O_2-_treated group than in the blank group. In contrast, the number of live cells was significantly higher in H_2_O_2_-treated neurons exposed to IBC extract. These findings suggest that IBC extract can effectively protect neuronal viability under in vitro-modeled ROS-generating conditions. Thus, IBC extract may provide neuroprotection against oxidative stress and can be safely applied as it does not induce cellular toxicity.

### 3.2. Effect of IBC Extract on Mitochondrial ROS Production Following H_2_O_2_-Induced Oxidative Stress Injury in Rat Primary Cortical Neurons

We assessed MitoSOX expression in cortical neurons treated with H_2_O_2_ to induce oxidative stress (Figure 2A). Our results show that MitoSOX expression in the cytoplasm was strongly and significantly higher in H_2_O_2_-treated cortical neurons than in blank neurons, suggesting a significant generation of mitochondrial ROS. When the treated cells were also exposed to the three IBC extract concentrations (5, 10, and 20 µg/mL), MitoSOX expression significantly decreased in a dose-dependent manner (Figure 2C). The flow cytometric assessment of MitoSOX positivity showed a similar trend to that observed in the confocal images (Figure 2B). MitoSOX positivity was substantially higher in H_2_O_2_-treated neurons than in those that had also been exposed to IBC extract (Figure 2D). In addition, we detected oxidized DCFDA using flow cytometry to confirm the level of intracellular ROS (Appendix A). CM-H_2_DCFDA (DCFDA) is a cell-permeable fluorogenic probe used as an indicator for intracellular ROS. DCFDA positivity revealed a dose-dependent decrease in ROS production by IBC with different concentrations in H_2_O_2_-treated neurons (Appendix A).

These results demonstrate that IBC extract protects cortical neurons from H_2_O_2_-induced oxidative stress by inhibiting mitochondrial ROS production in a dose-dependent manner.

### 3.3. Effect of IBC Extract on Mitochondrial Mass and Morphology Following H_2_O_2_-Induced Oxidative Stress Injury in Rat Primary Cortical Neurons

Observing changes in mitochondrial morphology is important for a better understanding of the complex pathological mechanism associated with oxidative stress. Previous studies have shown that the mitochondrial amount, shape, and distribution pattern are closely related to the maintenance of cell homeostasis when cells experience metabolic or environmental stresses [21]. In addition, correct mitochondrial balance is especially important for ensuring neuronal energy supplies, survival, and function. Ordinarily, mitochondria are evenly distributed in the cytoplasm, especially in their axons and dendrites. MitoTracker Red dye (MTR) has been used to visualize mitochondria and evaluate the mitochondrial membrane potential. The accumulation of MTR with mitochondria depends on the membrane potential of the mitochondria (Figure 3A). When H_2_O_2_ was applied to cortical neurons, the intensity of MTR decreased dramatically, and mitochondrial fragmentation increased. However, treatment with the IBC extract allowed the maintenance of high intensity of MTR throughout the cytoplasm. These findings indicate that IBC treatment led to a consistent increase in MTR fluorescence intensity, and could have contributed to increased mitochondrial membrane potential. Additionally, we also tried to determine the exact effect of IBC on mitochondrial mass in cortical neurons. We quantified the ratio between a target mitochondrial gene and a reference nuclear gene (mtDNA/nDNA) using quantitative real-time PCR (Figure 3B). The decrease in the mtDNA/nDNA ratio in the H_2_O_2_ group may reflect either a decrease in mitochondrial number/content or a depletion of mtDNA. Further, the ratio of mtDNA/nDNA was significantly higher in the IBC groups as compared to that in the H_2_O_2_ groups and may reflect either an increase in mitochondrial number/content or a higher mtDNA concentration. To further verify the number of mitochondria or the mitochondrial mass, we employed flow cytometry using MTR, which has been commonly used to assess mitochondrial mass (Figure 3C,D). MTR fluorescence was considerably lower in the H_2_O_2_ groups, in comparison with the blank group. In contrast, the percentage of MTR-positive cells increased significantly with IBC treatment. Further, the mitochondrial mass increased with IBC treatment in a dose-dependent manner in non-activated cortical neurons without H_2_O_2_ treatment (Appendix A, Appendix A).

In addition, we observed changes in mitochondrial shape in axons (Figure 3E). Some studies have reported that the typical morphology of a mitochondrion observed in an axon or dendrite is tubular in shape, with lengths ranging from a few to several tens of microns [22]. In the blank group, mitochondria were compactly distributed along the axon and generally appeared to exhibit a long oval appearance. However, damaged mitochondria were rarely observed along the axon and appeared highly fragmented after H_2_O_2_ treatment. In the IBC-treated group, immunocytochemical images revealed morphology similar to that of the blank group, with the mitochondria appearing elongated and ovular. Treatment with IBC induced dose-dependent increases in mitochondrial density. We further analyzed the expression level of genes associated with mitochondrial fission and fusion. Dynamic-related protein 1 (*Drp1*) and mitochondrial fission 1 (*Fis1*) critically mediate mitochondrial fission, whereas mitofusin-1 (*Mfn1*) and -2 (*Mfn2*) play key roles in mitochondrial fusion. Our results showed that *Drp1* and *Fis1* expression levels were significantly increased after H_2_O_2_ treatment in neurons, whereas their levels were well-regulated in a dose-dependent manner after IBC treatment (Figure 3F,G). In contrast, expression levels of fusion-related genes including *Mfn1* and *Mfn2* were significantly decreased in the H_2_O_2_ group. However, expression of these genes significantly and dose-dependently increased after application of IBC at different concentrations (Figure 3H,I). In addition, *Opa1* is the most widely employed gene for studying the mitochondrial dynamics with neuropathy and has well-known roles in both mitochondrial fusion and crista remodeling [23]. We further analyzed whether the IBC extract could mediate *Opa1* expression in H_2_O_2_ treated cortical neurons using real-time PCR. The *Opa1* gene decreased in the H_2_O_2_ group. However, the expression levels of *Opa1* were not significantly different between the H_2_O_2_ and IBC groups (Figure 3J).

These findings reveal that IBC induced changes in the expression of genes associated with mitochondrial morphology under H_2_O_2_-induced oxidative stress injury.

### 3.4. Effect of IBC Extract on H_2_O_2_-Induced Oxidative Stress-Related DNA Damage in Rat Primary Cortical Neurons

8-OHdG has been previously used as a sensitive biomarker of oxidative DNA damage and oxidative stress [24]. We used 8-OHdG evaluations to study oxidative DNA damage. First, we used immunochemistry to compare 8-OHdG expression in cells treated with H_2_O_2_ and those treated with H_2_O_2_ and IBC extract (Figure 4A). A large proportion of cells stained positive for 8-OHdG in the H_2_O_2_ group; however, the rate of 8-OHdG-positive cells significantly decreased in the IBC-treated groups. These findings indicate that IBC extract can effectively block oxidative DNA damage in cells treated with H_2_O_2_ in a dose-dependent manner (Figure 4B). We also evaluated oxidative damage in nDNA and mtDNA by measuring the level of 8-OHdG using a dot blot assay (Figure 4C,D). The 8-OHdG level in nDNA and mtDNA was significantly lower in the IBC group than in the H_2_O_2_ group, and such differences occurred in a dose-dependent manner. In parallel with the results of immunochemistry, the amount of 8-OHdG in nDNA and mtDNA was markedly decreased at 10 and 20 µg/mL IBC concentrations. These results suggest that IBC aids in preventing oxidative damage to nDNA and mtDNA.

### 3.5. Effect of IBC Extract on Mitochondrial Function Following H_2_O_2_-Induced Oxidative Stress Injury in Rat Primary Cortical Neurons

Normal mitochondrial function is critical for cellular growth, homeostasis, and survival, and ATP level through oxidative phosphorylation is considered a major function of the mitochondria. ATP is the source of energy in normal mammalian cells for survival, from both glycolysis and mitochondrial oxidative phosphorylation (OXPHOS). ATP is quantified by measuring the light produced through the addition of luciferase enzyme and luciferin using a luminometer. The quantity of light that is produced by the reaction is directly proportional to the presence of ATP in the sample. We first examined ATP level 20 min after treating H_2_O_2_-challenged neurons with IBC extract (5, 10, or 20 µg/mL; Figure 5A). H_2_O_2_ treatment induced a decrease in ATP level, but the IBC extract treatment induced a significant increase in ATP level in a dose-dependent manner when assessed at 2-min intervals over 20 min.

Furthermore, MMP is an essential component of energy storage, as it is used by ATP synthase to generate ATP. We further analyzed the MMP with JC-1 (Figure 5B,C), an MMP depolarization-specific dye. Healthy cells with a high rate of MMP JC-1 aggregates could be detected in the blank group, whereas JC-1 monomers were observed in H_2_O_2_-induced neurons. When various concentrations of the IBC extract were added to H_2_O_2_-treated neurons, the mitochondria exhibited increased formation of JC-1 aggregates, indicated by red fluorescence.

Next, we also confirmed the effect of IBC extract treatment on MMP using TMRM cell-permeant dye (Figure 6A). Neurons treated with H_2_O_2_ exhibited a marked reduction in TMRM signaling, whereas the IBC extract treatment increased such signaling (Figure 6B). We also found that IBC extract increased MMP in non-activated cortical neurons by flow cytometry. These values showed a significant dose-dependent difference from the blank group (Appendix A, Appendix A). Based on these findings, IBC extract treatment with up to 20 µg/mL protected mitochondrial function under in vitro-modeled ROS-generating conditions by maintaining mitochondrial membrane potential. In addition, we examined whether IBC enhances mitochondrial biogenesis-related genes (based on mRNA levels) using real-time PCR (Figure 6C–G). Previous studies have reported that heme oxygenase-1 (*HO-1*) is an antioxidant enzyme that is an important regulator of angiogenesis, mitochondria biogenesis, and neurogenesis by increasing the protein expression of *NRF-1*, *PGC1*, and *TFAM* [25]. The expression levels of mitochondrial biogenesis-related genes, including *HO-1*, *PGC1α*, and *TFAM*, were detected with qRT-PCR analysis. *PGC1α* gene expression was significantly upregulated in the IBC groups in a dose-dependent manner, whereas *HO-1* and *TFAM* showed a tendency to increase with the concentration but were only significantly upregulated in the 20 µg/mL IBC group (Figure 6C–E). In addition, nuclear respiratory factors, *Nrf-1* and *Nrf-2,* mediate expression of the electron transfer chain subunits encoded by the nuDNA or mtDNA and bind to the promoter sequence, thus initiating mtDNA transcription. *Nrf-1* binds to the specific promoter region and regulates the expression of *TFAM*, *TFB1M*, and *TFB2M*. In particular, *Nrf-2* regulates the expression of nuclear-encoded mitochondrial protein, TOMM20 (translocase of outer mitochondrial membrane 20), which plays a key role in maintaining mitochondrial biogenesis and function [26,27]. Therefore, *Nrf-1* and *Nrf-2* are well known for their role in the homeostasis of ROS by mediating the expression of antioxidant proteins through several mechanisms related to mitochondrial biogenesis. We further confirmed the *Nrf-1* and *Nrf-2* mRNA and protein expression levels in each group by real-time PCR and immunocytochemical staining. The mRNA levels of *Nrf-1* and *Nrf-2* increased significantly in the IBC groups than in the H_2_O_2_ group. In particular, we found that 20 µg/mL IBC was the most effective concentration for increasing the expression of these genes and facilitating mitochondrial biogenesis (Figure 6F,G). In addition, changes in the *Nrf-2* levels in cortical neurons were detected immunocytochemically in each group. The intensity of *Nrf-2* was higher in the IBC groups than in the H_2_O_2_ group. However, we did not find any differences in *Nrf-2* intensity between different IBC concentrations (Appendix A).

These data demonstrate that IBC enhances mitochondrial biogenesis in H_2_O_2_-treated neurons via *HO-1*, *PGC1*, *TFAM, Nrf-1, and Nrf-2* expression.

### 3.6. Effect of IBC Extract on Cell Death Following H_2_O_2_-Induced Oxidative Stress Injury of Rat Primary Cortical Neurons

Mitochondria play key roles in regulating cell death, which is triggered by mitochondrial CycC from the mitochondrial inter-membrane space into the cytoplasm [28]. When CycC is released due to mitochondrial swelling triggered by environmental stimuli, it can induce cell death through various processes such as apoptosis and necrosis [29]. We examined CycC expression using immunocytochemistry (Figure 7A,C). H_2_O_2_ treatment substantially increased CycC expression, while treatment with IBC extract significantly decreased such expression in a dose-dependent manner. Additionally, we used flow cytometry to confirm that neuronal death could occur via apoptosis or necrosis (Figure 7B,D). Approximately 30% of the cells died due to necrosis following H_2_O_2_ treatment (annexin V-/PI+). In contrast, treatment with IBC extract produced a gradual decrease in the rate of necrosis, reaching a minimum of approximately 17%. These findings further indicate that IBC extract can protect cortical neurons against H_2_O_2_-induced oxidative damage. Furthermore, we investigated the potential effect of IBC on mitophagy in H_2_O_2_-induced oxidative stress injury. Mitophagy is a mitochondria-specific process in which defective mitochondria are selectively degraded to promote mitochondrial turnover and maintain cellular homeostasis. The PINK1 and Parkin-mediated pathway is most widely accepted as the main mechanism underlying mitophagy [30]. We assessed the protein expression levels of PINK1 and parkin using Western blot analyses (Figure 7E,F). H_2_O_2_ treatment markedly decreased the protein expression level of PINK1 and parkin in cortical neurons, whereas treatment with IBC increased the amount of PINK1 and parkin protein when compared with that in the H_2_O_2_ group. In addition, the critical proteins in the PINK1–Parkin dependent pathway include the microtubule-associated protein 1 light chain 3β (MAP1LC3B), also known as LC3B, and p62, a classical receptor of autophagy. p62 recognizes poly-ubiquitinated proteins with autophagosomes by binding to the autophagy protein LC3. We assessed the expression of the autophagy markers LC3B and p62 by immunocytochemistry (Appendix A). The critical proteins in mitophagy, including LC3 and p62, showed elevated expression in the IBC groups.

Therefore, IBC plays a critical role in maintaining mitochondrial homeostasis by controlling these mitochondrial proteins, the PINK1–Parkin pathway.

### 3.7. Effect of IBC Extract on the Expression of Neurotrophic Factors Following H_2_O_2_-Induced Oxidative Stress Injury in Rat Primary Cortical Neurons

Neurotrophic factors provide neuroprotection and have traditionally been considered beneficial for the treatment of neurodegenerative diseases. BDNF and NGF are powerful neuroprotective factors associated with neuroprotection and neuroplasticity [31]. BDNF and NGF levels were analyzed via immunochemical staining of cortical neurons (Figure 8A,B). There was a significant decrease in BDNF and NGF expression when neurons were treated with H_2_O_2_. IBC extract treatment increased their expression in a dose-dependent manner (Figure 8C,D). These results suggest that treatment with the IBC extract increases BDNF and NGF expression within the cells, effectively stimulating the regrowth in injured axons. We also analyzed whether IBC promotes BDNF and NGF expression in cortical neurons without H_2_O_2_ stimulation using flow cytometry. BDNF and NGF expression showed a trend toward statistical significance following increased IBC concentrations in cortical neurons (Appendix A, Appendix A). When we investigated the changes in the expression of neurotrophin-related genes, such as *BDNF*, *NGF*, and *NT3*, using qRT-PCR analysis (Figure 8E–G), we found that the expression of *BDNF*, *NGF*, and *NT3* within the neurons was significantly higher in the 20 µg/mL IBC group than in the H_2_O_2_ group. However, it was not different between the two different IBC (5 and 10 µg/mL) groups and the H_2_O_2_ group.

### 3.8. Effect of IBC Extract on Synapse Formation and Stabilization Following H_2_O_2_-Induced Oxidative Stress Injury of Rat Primary Cortical Neurons

Synaptogenesis is important for synapse maturation, maintenance, and assembly of a new neural circuit. Syn is a useful biomarker of synapse formation. We analyzed its expression using immunocytochemistry (Figure 9A). Interestingly, treatment with IBC extract-enhanced Syn expression following H_2_O_2_-induced injury, whereas its expression appeared to be absent in the H_2_O_2_ alone group. The quantification revealed that the relatively high intensity of Syn detected in the IBC groups was dose-dependent (Figure 9B). Furthermore, we compared the length of the longest axon following the various treatments at 7 DIV (Figure 9C). These results demonstrate that treatment with IBC extract enhances axonal elongation after the H_2_O_2_ challenge, significantly increasing the longest axonal length. The results also revealed that an incremental increase in the IBC extract dose significantly enhanced synapse formation following H_2_O_2_-induced oxidative stress injury in rat primary cortical neurons. qRT-PCR analysis was performed to detect changes in neurite outgrowth-related gene expression after three different concentrations of IBC treatment in H_2_O_2_-treated neurons. The regeneration-related genes, growth-associated protein (*GAP43*), and neurofilament 200-kDa (*NF200*), were upregulated in the IBC groups compared to in the H_2_O_2_ group but were similar in two doses of IBC (5 and 10 µg/mL, Figure 9D,E).

Here, we summarize the therapeutic role of IBC through mitochondrial ROS scavenging, suppressing oxidative stress and cell death, and enhancing neurite outgrowth and synaptogenesis, as illustrated in Figure 10.

## 4. Discussion

Mitochondria are key regulators of cell survival and death and have been associated with neurodegenerative diseases characterized by progressive failure of brain systems due to selective neuronal vulnerability. Previous studies have shown that the number, morphology, and distribution of mitochondria in neurons are critical factors affecting their growth, survival, and homeostasis [21]. Mitochondrial dysfunction occurs in the early stages of neurodegenerative diseases and has been highlighted as a potential cause of such disorders. Additionally, mitochondrial-derived oxidative stress is important in the pathogenesis of metabolic syndrome, rheumatoid arthritis, and neuropathies [32]. However, few studies have focused on the exact cause of ROS-mediated cellular dysfunction, the associated metabolic processes, and the basic mechanisms underlying the efficacy of plant-based drugs. The concentration of antioxidative substances obtained from various natural products, such as plant leaves, fruits, and extracts of animals and plants, seems crucial in this regard. Our findings verified that mitochondrial-induced ROS increases in cortical neurons following H_2_O_2_-induced injury, resulting in oxidative DNA damage and changes in the regulation of mitochondrial biogenesis. While several studies have already reported on the role of mitochondria in cell death regulation, this study aimed to verify the neuroprotective effect of IBC flower extract—an excellent natural antioxidative substance already in use in traditional herbal medicine—based on the association between mitochondrial roles and ROS metabolism in cortical neurons. Our investigation of the mechanism of action, efficacy, and safety of IBC extract establishes an objective and scientific basis for its use in the treatment of neurodegenerative diseases. Indeed, our findings demonstrate that IBC extract contributes to the improvement of mitochondrial biogenesis following oxidative stress, facilitating the oxygen supply required for energy metabolism, thereby aiding in increases in ATP level. Furthermore, the extract inhibits the activation of oxidative stress-related factors, confirming its neuroprotective effect by improving the oxidative stress environment. Interestingly, our results also indicated that IBC extract increases BDNF and NGF expression and induces nerve growth by reducing Syn loss, a major synaptic vesicle protein. However, further studies are required to examine the therapeutic effect of IBC extract in models of neurological disease. In addition, future studies should aim to determine whether specific genes and proteins act on and contribute to the morphological and functional changes in the mitochondria after injury.

## 5. Conclusions

Our findings support the notion that the neuroprotective effect of the IBC extract on cortical neurons is associated both with a decrease in mitochondrial-derived ROS and an increase in mitochondrial-derived ATP level. Moreover, the effect of IBC against H_2_O_2_-induced injury through increased MMP and mitochondrial biogenesis related genes were confirmed. Our results suggest that the IBC extract can be applied in the treatment of neurological diseases. Further, our results can serve as an important data resource when determining a standard treatment direction for oxidative stress regulation using several existing natural product-based antioxidizing agents.

## Figures and Tables

**Figure 1 antioxidants-10-00375-f001:**
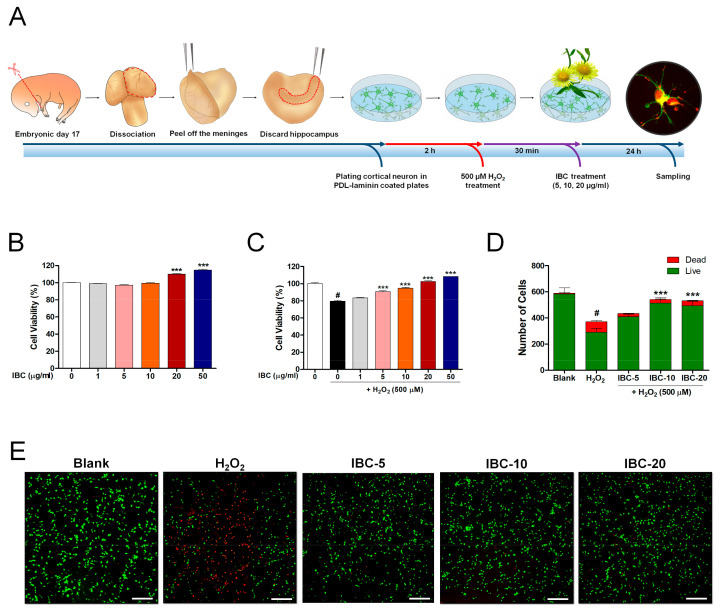
*Inula britannica* var. chinensis (IBC) extract potentially protects against H_2_O_2_-induced oxidative stress injury in rat primary cortical neurons. (**A**) Schematic drawing of the experimental protocol of a primary cortical neuron. (**B**,**C**) CCK-8 results of cortical neurons treated with various concentrations of IBC for 24 h with (n = 6) and without exposure to hydrogen peroxide stress (n = 4). (**D**) Quantification of the number of live and dead cells for the neurotoxicity assay (n = 5). (**E**) Cell viability was imaged with confocal microscopy using live/dead imaging assay kit (live cells, green; dead cells, red). White scale bar = 200 µm. Data are expressed as the means ± SEM. Significant differences indicated as # *p* < 0.001 compared vs. the blank group, *** *p* < 0.001 vs. the H_2_O_2_ group were analyzed via one-way ANOVA with Tukey’s post hoc test.

**Figure 2 antioxidants-10-00375-f002:**
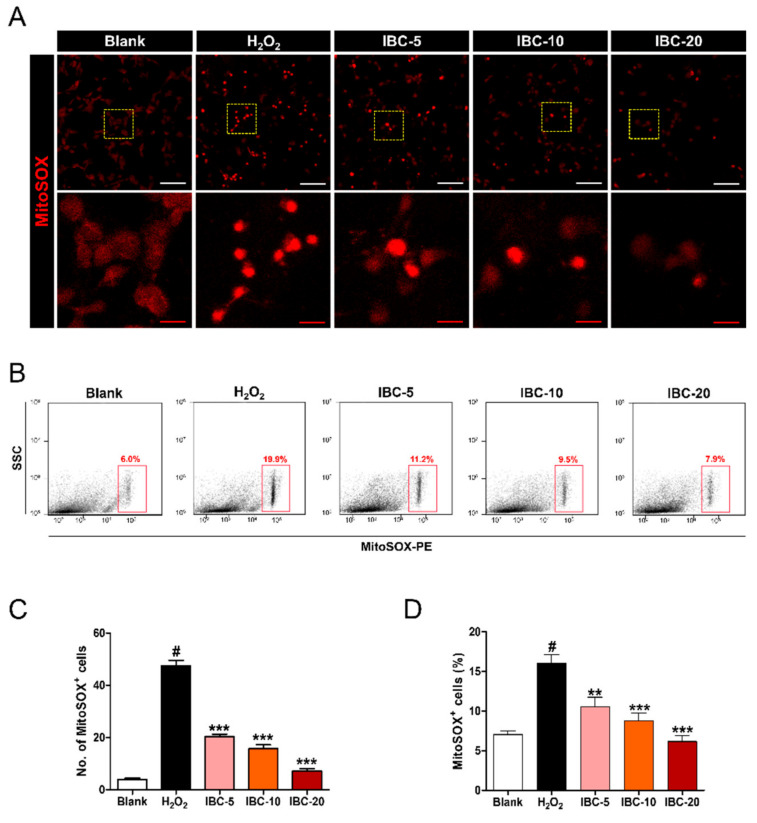
Effect of IBC extract on mitochondrial reactive oxygen species (ROS) production following H_2_O_2_-induced oxidative stress injury in rat primary cortical neurons. (**A**) Representative images of MitoSOX-stained neurons after IBC application in H_2_O_2_-treated neurons. White scale bar = 50 µm, Red scale bar = 10 µm. (**B**) Flow cytometric plots of cortical neurons stained with MitoSOX. side scattered light (SSC) (**C**) Quantification of the number of MitoSOX-labeled cells in each group (n = 9). (**D**) The percentage of positive cells for MitoSOX from flow cytometric immunofluorescence (n = 5). Data are expressed as the means ± SEM. Significant differences indicated as # *p* < 0.001 compared vs. the blank group, ** *p* < 0.01 and *** *p* < 0.001 vs. the H_2_O_2_ group were analyzed via one-way ANOVA with Tukey’s post hoc test.

**Figure 3 antioxidants-10-00375-f003:**
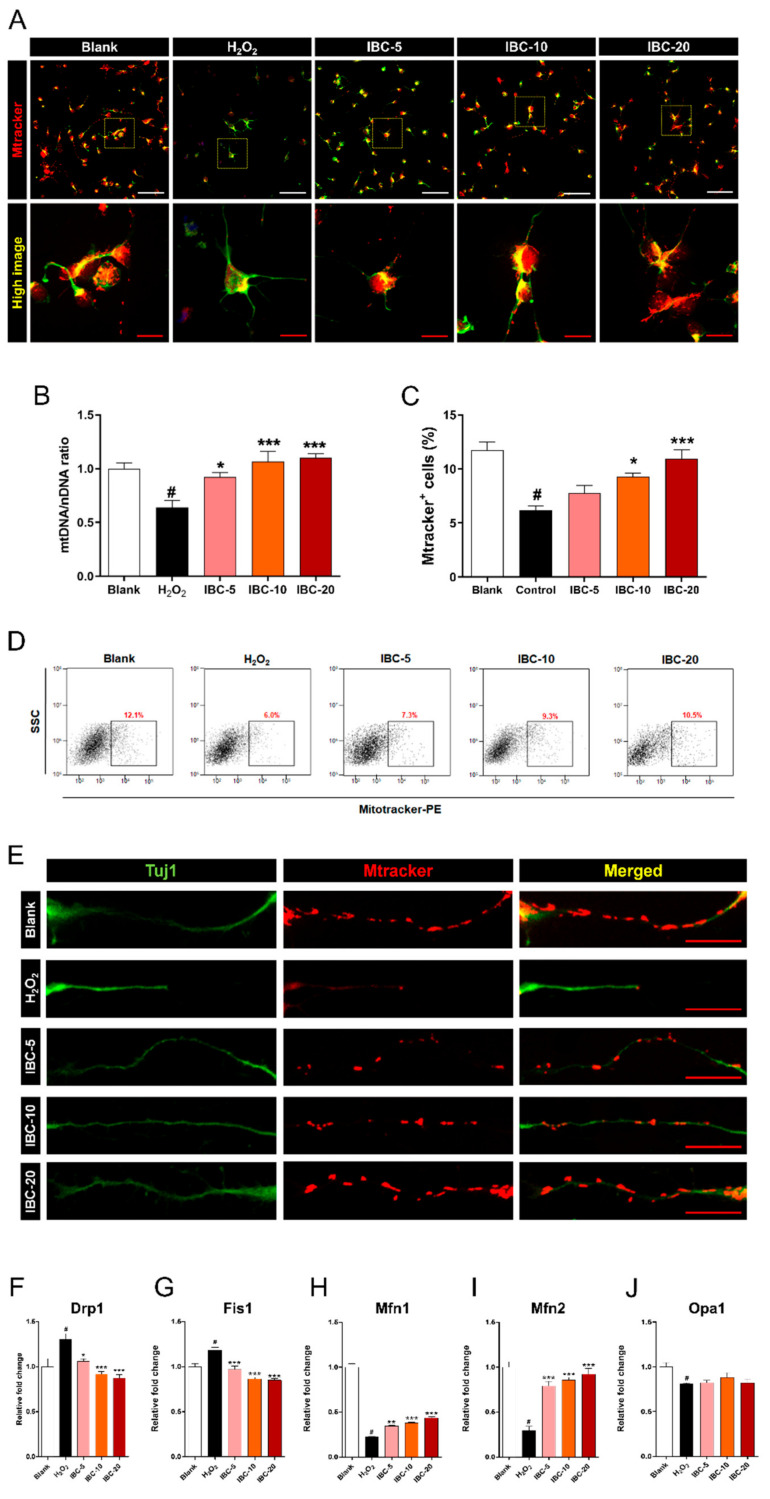
Effect of IBC extract on mitochondrial amount and morphology following H_2_O_2_-induced oxidative stress injury in rat primary cortical neurons. (**A**) Representative images showing expression of Mitotracker marker in IBC-treated neurons. White scale bar = 50 µm, Red scale bar = 10 µm. (**B**) Analysis of mtDNA/nDNA ratio by quantitative real-time PCR (n = 6). (**C**) Quantification of Mitotracker-positive cells at the single-cell level by flow cytometry (n = 4). (**D**) Representative flow cytometric dot plots graphs showing Mitotracker expression. (**E**) High magnified images showing mitochondria within axons. Red scale bar = 10 µm. (**F**–**J**) Expression of mitochondrial fission and fusion-related genes by qRT-PCR (n = 6); (**F**) *Drp1*, (**G**) *Fis1*, (**H**) *Mfn1*, (**I**) *Mfn2 and* (**J**) *Opa1*. Data are expressed as the means ± SEM. Significant differences indicated as # *p* < 0.001 compared vs. the blank group, * *p* < 0.05, ** *p* < 0.01 and *** *p* < 0.001 vs. the H_2_O_2_ group were analyzed via one-way ANOVA with Tukey’s post hoc test.

**Figure 4 antioxidants-10-00375-f004:**
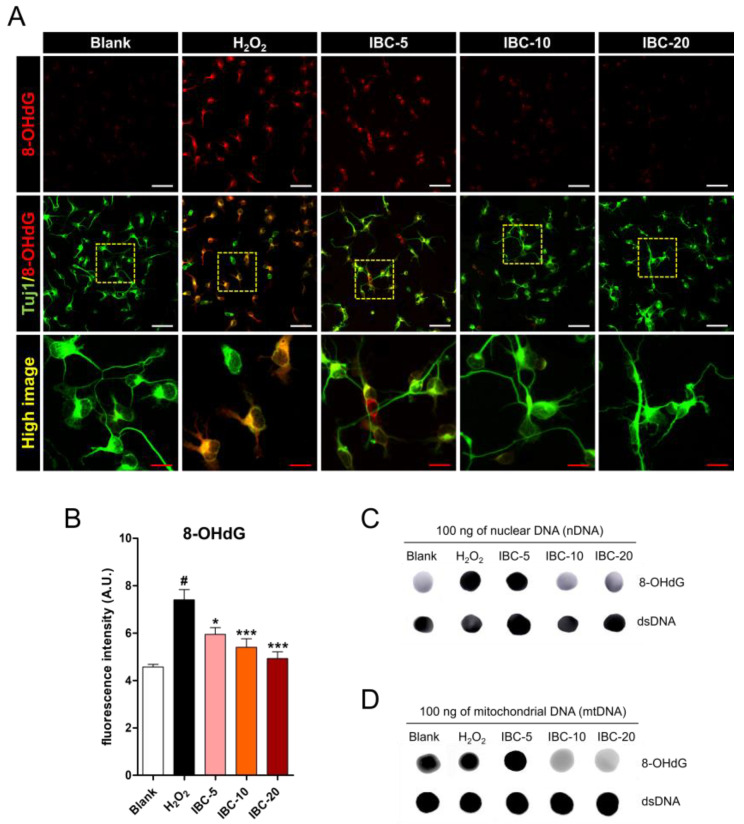
Effect of IBC extract on H_2_O_2_-induced oxidative stress-related DNA damage in rat primary cortical neurons. (**A**,**B**) Representative images (**A**) and quantitative analysis (**B**) showing the degree of oxidation in DNA stained with 8-OHdG for all five groups (n = 7). White scale bar = 50 µm, Red scale bar = 15 µm. (**C**,**D**) Representative dot blot images of 8-OHdG levels in nuclear DNA (nDNA) and mitochondrial DNA (mtDNA). Data are expressed as the means ± SEM. Significant differences indicated as # *p* < 0.001 compared vs. the blank group, * *p* < 0.05 and *** *p* < 0.001 vs. the H_2_O_2_ group were analyzed via one-way ANOVA with Tukey’s post hoc test.

**Figure 5 antioxidants-10-00375-f005:**
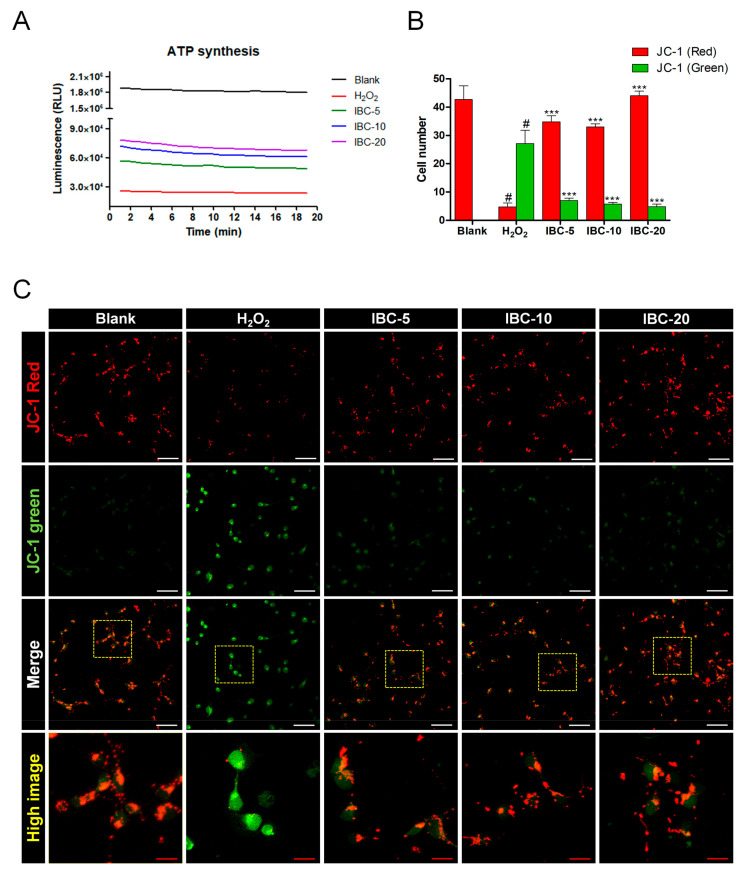
Effect of IBC extract on mitochondrial function following H_2_O_2_-induced oxidative stress injury in rat primary cortical neurons. (**A**) Quantifying cellular ATP level in the mitochondria. (**B**,**C**) Quantitative analysis (**B**) and representative images (n = 4) (**C**) of JC-1-stained mitochondria for assessing mitochondrial membrane potential (ΔΨ m). White scale bar = 50 µm, Red scale bar = 15 µm. Data are expressed as the means ± SEM. Significant differences indicated as # *p* < 0.001 compared vs. the blank group, *** *p* < 0.001 vs. the H_2_O_2_ group were analyzed via one-way ANOVA with Tukey’s post hoc test.

**Figure 6 antioxidants-10-00375-f006:**
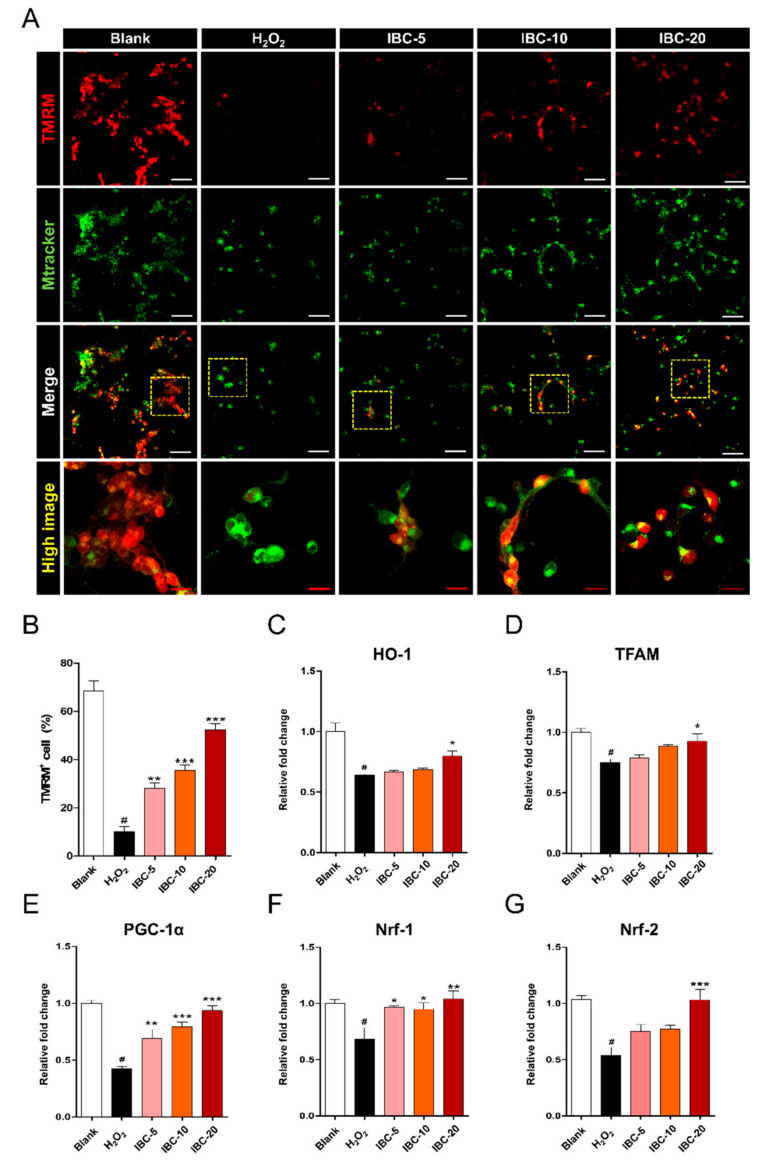
Effect of IBC extract on mitochondrial biogenesis following H_2_O_2_-induced oxidative stress injury in rat primary cortical neurons. (**A**) Representative images of TMRM stained neurons in each group. White scale bar = 50 µm, Red scale bar = 15 µm. (**B**) Quantification of the relative percentage of the total number of TMRM+ cells in cortical neurons (n = 6). (**C**–**G**) Expression of genes related to mitochondrial biogenesis following H_2_O_2_ treatment and the application of varying doses of IBC for 24 h (n = 6); (**C**) *HO-1*, (**D**) *TFAM*, (**E**) *PGC-1α*, (**F**) *Nrf-1* and (**G**) *Nrf-2*. Data are expressed as the means ± SEM. Significant differences indicated as # *p* < 0.001 compared vs. the blank group, * *p* < 0.05, ** *p* < 0.01 and *** *p* < 0.001 vs. the H_2_O_2_ group were analyzed via one-way ANOVA with Tukey’s post hoc test.

**Figure 7 antioxidants-10-00375-f007:**
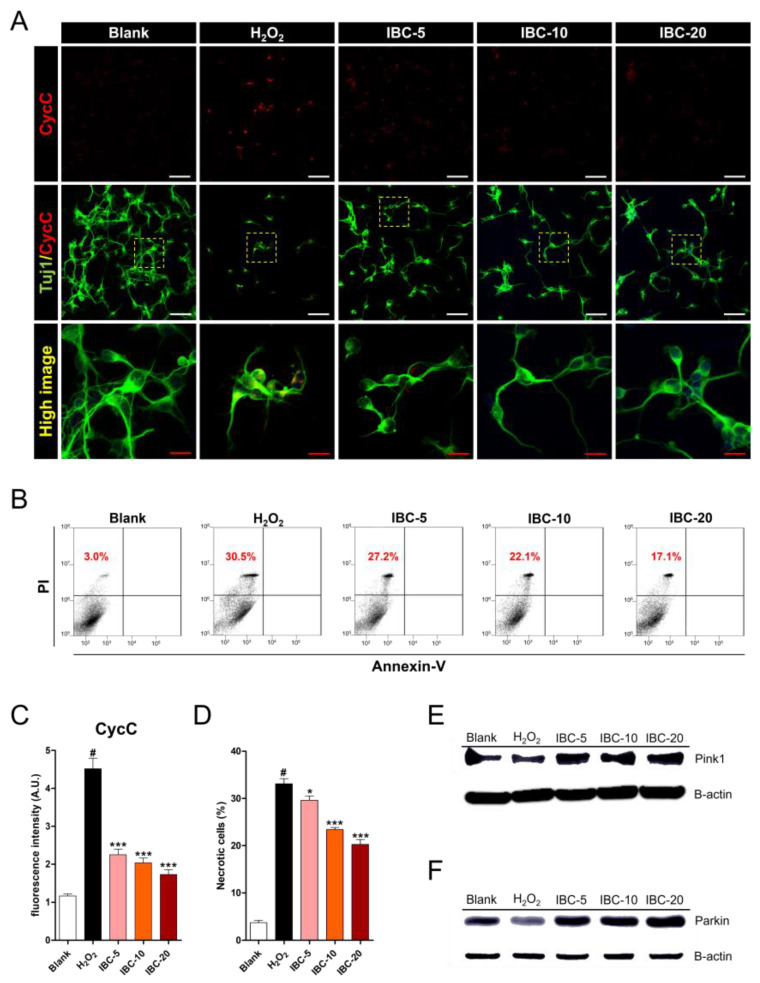
Effect of IBC extract on cell death following H_2_O_2_-induced oxidative stress injury of rat primary cortical neurons. (**A**) Representative images of low and high anti-cytochrome c (CycC) staining. White scale bar = 50 µm, Red scale bar = 15 µm. (**B**) Flow cytometric dot plots showing annexin V (X-axis) and propidium iodide (Y-axis) staining of cortical neurons. (**C**) The relative fluorescence intensity of anti-CycC on each low magnified image in five groups (n = 6). (**D**) The percentage of necrotic cells for annexin V-/PI+ from flow cytometric immunofluorescence (n = 5). (**E**,**F**) Western blot analysis of pink1 and parkin in IBC-treated neurons with H_2_O_2_ exposure. Data are expressed as the means ± SEM. Significant differences indicated as # *p* < 0.001 compared vs. the blank group, * *p* < 0.05 and *** *p* < 0.001 vs. the H_2_O_2_ group were analyzed via one-way ANOVA with Tukey’s post hoc test.

**Figure 8 antioxidants-10-00375-f008:**
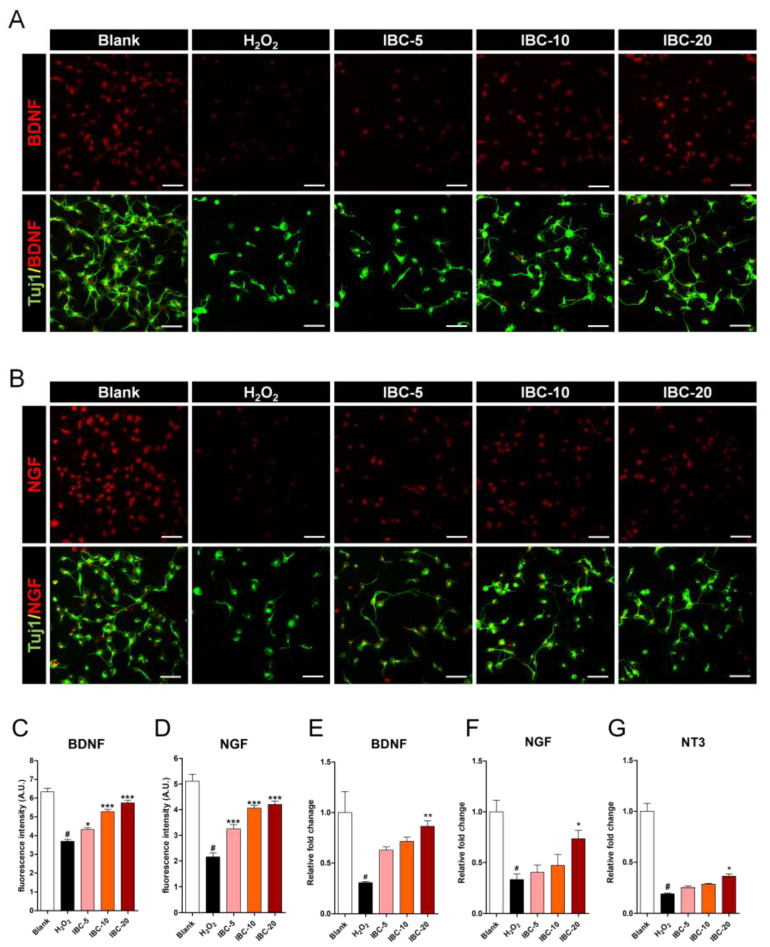
Effect of IBC extract on the expression of neurotrophic factors following H_2_O_2_-induced oxidative stress injury in rat primary cortical neurons. (**A**,**B**) Representative images of anti-BDNF (**A**) and NGF (**B**) staining. White scale bar = 50 µm. (**C**,**D**) Quantifying mean fluorescence intensities of BDNF (**C**) and NGF (**D**) in each group (n = 7). (**E**–**G**) Gene expression associated with neurotrophic factors in cortical neurons using qRT-PCR (n = 6); (**E**) *BDNF*, (**F**) *NGF* and (**G**) *NT3*. Data are expressed as the means ± SEM. Significant differences indicated as # *p* < 0.001 compared vs. the blank group, * *p* < 0.05, ** *p* < 0.01 and *** *p* < 0.001 vs. the H_2_O_2_ group were analyzed via one-way ANOVA with Tukey’s post hoc test.

**Figure 9 antioxidants-10-00375-f009:**
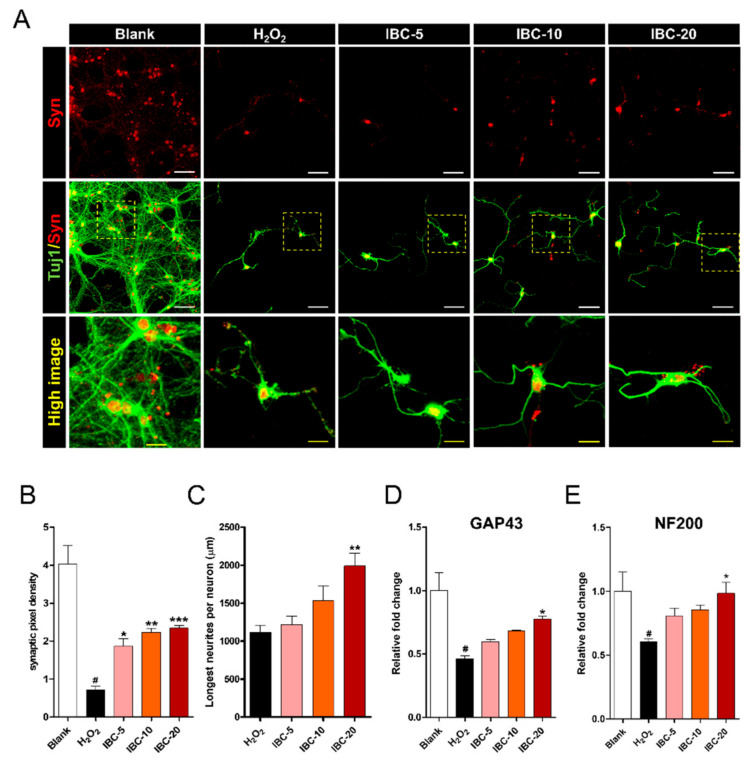
Effect of IBC extract on synapse formation and stabilization following H_2_O_2_-induced oxidative stress injury of rat primary cortical neurons. (**A**) Representative images of low and high anti-synaptophysin (Syn) staining at 7 DIV (day in vitro). White scale bar = 50 µm, Yellow scale bar = 15 µm. (**B**,**C**) Quantification of synaptic densities (**B**) and the length of the longest neurite (**C**) determined by Tuj1 immunoreactivity in neurons at 7 DIV (n = 7). (**D**,**E**) Gene expression associated with neuronal growth in cortical neurons, assessed using qRT-PCR (n = 6); (**D**) *GAP43* and (**E**) *NF200*. Data are expressed as the means ± SEM. Significant differences indicated as # *p* < 0.001 compared vs. the blank group, * *p* < 0.05, ** *p* < 0.01 and *** *p* < 0.001 vs. the H_2_O_2_ group were analyzed via one-way ANOVA with Tukey’s post hoc test.

**Figure 10 antioxidants-10-00375-f010:**
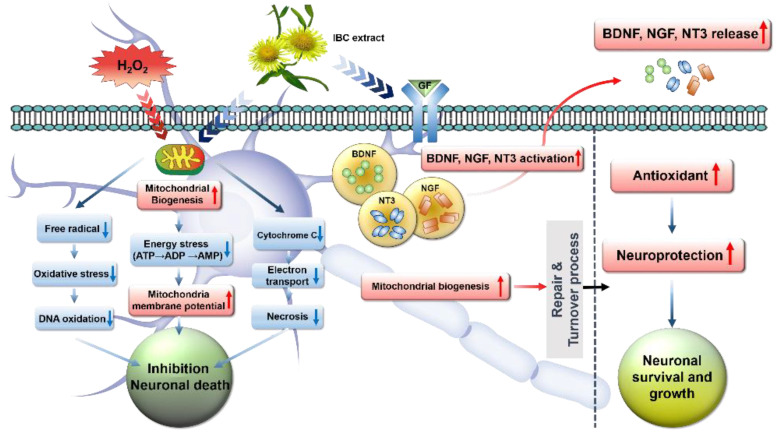
Schematic illustration showing the cascade of events involved in mitochondrial dysfunction after H_2_O_2_ treatment and the therapeutic regulation of IBC extract with varying concentrations through ROS scavenging, suppressing oxidative stress and cell death, enhancing mitochondrial function and neuronal cell growth, and consequently axonal regeneration.

**Table 1 antioxidants-10-00375-t001:** Primer sequences used for real-time PCR analysis.

Gene	5′–3′	Primer Sequence
*Drp1*	Forward	AGGTTGCCCGTGACAAATGA
	Reverse	CACAGGCATCAGCAAAGTCG
*Fis1*	Forward	GCCTGGTTCGAAGCAAATAC
	Reverse	CACGGCCAGGTAGAAGACAT
*Mfn1*	Forward	TTGCCACAAGCTGTGTTCGG
	Reverse	TCTAGGGACCTGAAAGATGGGC
*Mfn2*	Forward	GGGGCCTACATCCAAGAGAG
	Reverse	GCAGAACTTTGTCCCAGAGC
*Opa1*	Forward	ATCATCTGCCACGGGTTGTT
	Reverse	GAGAGCGCGTCATCATCTCA
*HO-1*	Forward	CCCACCAAGTTCAAACAGCTC
	Reverse	AGGAAGGCGGTCTTAGCCTC
*TFAM*	Forward	CCAAAAAGACCTCGTTCAGC
	Reverse	CCATCTGCTCTTCCCAAGAC
*PGC-1α*	Forward	CAGGAACAGCAGCAGAGACA
	Reverse	GTTAGGCCTGCAGTTCCAGA
*Nrf-1*	Forward	TAGCCCATCTCGTACCATCAC
	Reverse	TTTGTTCCACCTCTCCATCAG
*Nrf-2*	Forward	GATCTGTCAGCTACTCCCAG
	Reverse	GCAAGCGACTCATGGTCATC
*BDNF*	Forward	CTTGGAGAAGGAAACCGCCT
	Reverse	GTCCACACAAAGCTCTCGGA
*NGF*	Forward	CCAAGGACGCAGCTTTCTATC
	Reverse	CTGTGTCAAGGGAATGCTGAAG
*NT3*	Forward	CCGACAAGTCCTCAGCCATT
	Reverse	CAGTGCTCGGACGTAGGTTT
*GAP43*	Forward	TGCCCTTTCTCAGATCCACT
	Reverse	TTGCCACACAGAGAGAGAGG
*NF200*	Forward	AACACCACTTAGATGGCGGG
	Reverse	ACGTGGAGCGTTCAGCAATA
*GAPDH*	Forward	CCCCCAATGTATCCGTTGTG
	Reverse	TAGCCCAGGATGCCCTTTAGT

*BDNF*: brain-derived neurotrophic factor; *Drp1:* dynamin-related protein 1; *Fis1*: mitochondrial fission 1; *GAP43*: growth-associated protein 43; *GAPDH*: Glyceraldehyde 3-phosphate dehydrogenase; *HO-1*: heme oxygenase 1; *Mfn1*: mitofusin 1; *Mfn2*: mitofusin 2; *NF200*: neurofilament 200; *NGF*: nerve growth factor; *Nrf-1*: nuclear respiratory factor-1; *Nrf-2*: nuclear respiratory factor-2; *NT3*: neurotrophin 3; *Opa1*: OPA1 mitochondrial dynamin like GTPase; *PGC-1α*: peroxisome proliferator-activated receptor gamma coactivator 1-alpha; *TFAM*: mitochondrial transcription factor A.

## Data Availability

The data presented in this study are available on request from the corresponding author.

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
