# Peer review of "Neurotherapeutic Effect of Inula britannica var. Chinensis against H2O2-Induced Oxidative Stress and Mitochondrial Dysfunction in Cortical Neurons"

_antioxidants, 2021, doi:10.3390/antiox10030375_

Round 1

Reviewer 1 Report

In this manuscript Jin Young Hong and collaborators described in vitro the antioxidant property and the neuroprotective effects of  Inula britannica var. chinensis (IBC).

In the present study, authors investigated the effects of different concentrations of IBC extract  on cortical neurons using a hydrogen peroxide (H2O2)-induced injury model.

The manuscript is well written, with a good level of novelty/impact for this journal.

However I think that actually this manuscript presents some weakness that could be implemented in order to improve the quality of this manuscript.

In particular I have some concerns related different aspects of the methodology and interpretation of the results:

Methods

2.2. Preparation of I. britannica var. chinensis (IBC) extract

The IBC extract was prepared as follows: 300 g of IBC was heated to 105°C in 3 L of

distilled water for 3 h. After cooling at -20°C for 30 min….

Several papers report a different protocol for the preparation of the IBC extract, please, justify the rationale of this protocol or provide a reference.

Results

3.2. Effect of IBC extract on mitochondrial ROS production following H2O2-induced oxidative

stress injury in rat primary cortical neurons

After the MitoSox experiement, authors concluded that these results demonstrate that IBC extract protects cortical neurons from H2O2-induced oxidative stress by  inhibiting mitochondrial ROS production in a dose-dependent manner.

Well, the analysis of the Mitosox fluorescence represent an indication, however authors should put more effort to support their conclusion, for example authors could analyse  the anti-ROS defense system, via western blot or immuncytochemistry (Nrf2 activation, SOD2 levels, Aconitase activity).

3.3. Effect of IBC extract on mitochondrial number and morphology following H2O2-induced oxidative stress injury in rat primary cortical neurons

Authorse wrote: we observed mitochondrial changes by LABELLING  mitochondria in live cells with the MitoTracker dye, as shown in Figure 3a. Ordinarily, mitochondria are evenly distributed in the cytoplasm, especially in their axons and dendrites. When H2O2 was applied to cortical neurons, the NUMBER OF MITOCHONDRIA DRAMATICALLY DECREASED and mitochondrial fragmentation increased. However, treatment with IBC extract allowed for the maintenance of a high number of mitochondria throughout the cytoplasm. Therefore, these findings indicate that IBC treatment attenuated mitochondrial loss and pathological changes in neurons in a dose dependent manner.

In this context, the MitoTraker dye don’t represent the right strategy to quantify the mitochondrial mass.

First, the CCCP uncopling should be used as internal control to asses the MMP lost.

Second, the MitoTraker staining depends by the mitochondrial membrane potential. If the H2O2 induce a lost  of mitochondrial  membrane potential,  the reduced signal could depend by the reduced amount of Mitotraker that enter inside the mitochondria and so it does not indicate that there is a reduction of the mitochondrial number.

To better understand this point, authors should perform:

  1. a staining with anti-Tom20, in order to bypass the mitochondrial membrane potential problem
  2. a WB experiment quantifing the amount of some mitochondrial proteins (i.e. VDAC1; OXPHOS cocktail etc).
  3. quantify the mtDNA

If all these results go in the same direction, authors could assert that the H2O2 induce a reduction of the number of mitochondria and on the contrary IBC treatment increase the mitochondrial mass.

Moreover, in the same section, authors analyse the expression of proteins related to mitochodrial dynamics (Drp1, Fis1, MFN1 and MFN2), what about the well known profusion protein Opa1 ?

3.5. Effect of IBC extract on mitochondrial function following H2O2-induced oxidative stress  injury in rat primary cortical neurons

Previous studies have reported that heme oxygenase-1 (HO-1) is an antioxidant enzyme that is an important regulator of angiogenesis, mitochondria biogenesis, and neurogenesis by increasing the protein expression of NRF1, PGC1, and TFAM [20]. The expression levels of mitochondrial biogenesis-related genes, including HO-1, PGC1, and TFAM, were significantly increased in the IBC groups when compared to levels in the H2O2 group. These data demonstrate that IBC enhances mitochondrial biogenesis in H2O2-treated neurons via HO- 1, PGC1, and TFAM expression.

Also in this case, authors shold evaluate other markers of mitochondrial biogenesis:

  • Transcription levels of NRF1-NRF2
  • mitochondrial proteins amount (i.e. respiratory subunits)
  • mtDNA

3.6. Effect of IBC extract on mitochondrial-initiated apoptosis following H2O2-induced oxidative

stress injury of rat primary cortical neurons

In this section, the WB experiment performed to explain the mitophagy should  be implemented with teh assesment of the poliubiquitin status or analysing the LC3-P62 system.

Minor

There are some typos:

Lines 167-168

Institutes of Health).induction of cellular apoptosis. JC-1

Line 170: … high mitochondrial membrane 2.8.

Immunocytochemistry…..

Author Response

Dear Reviewer

We thank the editor and the reviewers for their excellent and constructive comments, which clearly

helped to improve the quality of this manuscript. We have performed additional experiments as

detailed below, thereby addressing the issues raised by the reviewers. We are pleased to provide the following point-to-point reply. Appropriated changes, suggested by the Reviewers, has been introduced to the manuscript (highlighted within the manuscript). We hope that our manuscript will be acceptable for publication in Antioxidants

Kind regards,

In-Hyuk Ha, M.D., Ph.D.

Reviewer 2 Report

In order to develop natural product-based therapies that utilize mitochondrial functions for the treatment of neurological diseases, in this study attention is paid to the effects of Inula britannica var. chinensis (IBC) extract on primary cortical cultures from rat embryos.  

The researchers analyse molecular parameters concerning the morphology and functional state of mitochondria, taking into consideration that the mitochondria play important physiological roles in maintaining cellular homeostasis, such as cell growth, cell cycle regulation, production of reactive oxygen species (ROS), and cell death. Moreover, the mithocondria regulate the synthesis of neurotransmitters and therefore all the processes of brain development, neuroplasticity and neurogenesis.

The methods chosen fall within the physiological and histochemical techniques. Real-time PCR, Western blotting, flow cytometry are also applied.

I believe the research was well planned and the results well documented. The schematic illustrations and figures are more than satisfactory. The interpretation of the results is linear and the discussion/conclusion is clear and concise.

Author Response

Dear reviewer

We thank the editor and the reviewers for their excellent and constructive comments, which clearly helped to improve the quality of this manuscript.

We have performed additional experiments asdetailed below, thereby addressing the issues raised by the reviewers.

We are pleased to provide the following point-to-point reply. Appropriated changes, suggested by the Reviewers, has been introduced to the manuscript (highlighted within the manuscript).

We hope that our manuscript will be acceptable for publication in Antioxidants

Kind regards,

In-Hyuk Ha, M.D., Ph.D.

Reviewer 3 Report

The manuscript describes the neurotherapeutic effect of a natural substance, the Inula britanica var. chinensis (IBC), using for this purpose a in vitro model of primary cortical neurons. In particular, authors use H202 treatment to induce the oxydative stress.

The manuscript is very well organized, methods are clearly described and the results presented in a explicit way. Nevertheless, there are some parts that should be improved.

Major

  • Authors should explain the reason why they use cortical primary neurons after only 2 h of their preparation? At this time, isolated and seeded cells are not yet developed and they are not mature physiologically, metabolically, morphologically…when using primary neuron cultures, cells let some days in culture before performing any kind of experiment/test in order to let them differentiate and re-adquire their original phenotype.
  • When setting the [H202] optimal concentration to induce the oxidative stress insult, have dose-response and / or time-course experiments been performed in order to ascertain the optimal dose and time of H202 exposition in their ouwn model?

Minor

  • Do authors thought about studying the effects of IBC in the context of other neurodegenerative pathologies such as e.g. hypoxic/ischemic injury, for which the same or other in vitro models can be used?
  • Through the manuscript, there is not any information regarding the number of biological and technical replicates used to perform this study. They should be specified.
  • Pg 3, line 99, include in brackets “(see the schematic drawing/experimental desing… in Fig. 1A)” at the end of the sentence.
  • Since the cells used to perform the sudies can from the same treatment scheme, it should be more clear if authors refer to the scheme in figure 1A. This statement is not always clear:
    • For instance, in pg 3, lines from 117 to 119. It might be changed with “Primary cortical neurons seeded on glass co-verslips in 24-well plates and prepared as described in Fig.1A were washed….”
    • Idem, pg 3, at lines 138-140. “…medium. Cells seeded in opaque-walled 96-well plates at a density of 2 ×104 cells/well were prepared as described in Fig.1A. The CellTiterGlo…”
    • Pg 4, lines 156-157, “ …protocol. Primary cortical neurons were seeded onto 24 well plates and prepared as described (Fig.1A). The cells were then stained…”
    • Pg 5, lines 239-240, “Briefly, the cells prepared as already described (Fig.1A), were collected and…”
  • Pg 4, line 172, there is something wrong in the paragraph lines 168-172. “…Institutes of Health). in induction of cellular apoptosis. JC-1, a sensitive mitochondrial dye, was used to evaluate the ΔΨm. JC-1 can aggregates in the mitochondrial matrix to yield red uorescence (λem = 590 nm) at high mitochondrial membrane…”
  • Pg 4, line 170, 2.8. Immunocytochemistry…, star a new paragraph.
  • Pg 4, line 172, “Immunocytochemistry was performed with the indicated antibodies in the H2O2-activated cortical neurons”, this is rendundant since antibodies are specified bellow.
  • Pg 4, line 182, no washes performed between primary and secondary antibodies incubation?, please, specify buffer and the conditions (time, temperature) of washes.
  • Pg 5, line 200, specify the composition of the blocking solution, that is the buffer used to prepare the 5% skim milk.
  • Pg 6, line 233, anti- PINK1 (PTEN-induced putative kinase 1), the abbreviation should be described the first time it appears.
  • Figures 2B, Supplementary Figure S1 A, C and Supplementary Figure S2 A, C, it should be specified what is represented int the Y axis.
  • 3 C-D-E-F, Fig. 6 C-D-E, Fig. 8 E-F-G, Fig. 9 D-E-F, represent graphs of relative gene expression. Two comment:
    • the name of the genes should be in italic;
    • what is the Reference group in each of these graphs?, Reference group should have a value of 1. Please, explain this item.

  • All graphs included in the manuscript in which the resuls of the immunostaining are included should specified what is represented in the Y axis. “Intensity” is not a valid paramether. It should be specify “fluorescence intensity A.U.”
  • 9 B, Y axis “synaptic density” is should be indicated what has measured to study the synaptic density, e.g. fluorescence intensity A.U.
  • Caption of Supplementary Figure 1, there is a mistake, “B)”has been written twice.

Author Response

(The authors gave the same response as above.)

Reviewer 4 Report

In this manuscript the authors look at the protective effects of an extract from Inula Britannica var. chinensis (IBC) on H2O2-mediated toxicity in primary rat cortical neurons. The study is clearly written and quite comprehensive. However, the authors never justify their use of this model. Why is H2O2 treatment of primary cortical neurons a good model for the damage that occurs in neurodegenerative diseases? Furthermore, a number of the results are either incorrectly interpreted or over-interpreted which significantly reduces enthusiasm for the study. These concerns are listed below.

  1. Figure 3 shows results for Mitotracker Red. However, Mitotracker Red accumulation in mitochondria is dependent on the mitochondrial membrane potential. Since the authors show in Figure 5 that H2O2 decreases the mitochondrial membrane potential, they cannot say that the decrease in staining with Mitotracker Red is due to a loss of mitochondria rather than a loss of mitochondrial membrane potential

  1. In Figure 5, how do the authors know that the ATP is coming from mitochondria? ATP is also generated by glycolysis which could also be impacted by H2O2. Also, it seems to me that Fig. 5A shows the levels of ATP not ATP production.

  1. line 404: The authors state that IBC maintains mitochondrial respiration but they never check this.

  1. Figure 6: The authors say that the levels of markers of mitochondrial biogenesis were significantly increased in the IBC groups when compared to the H2O2 group but this is only true for the highest IBC dose for HO1 and TFAM.

  1. Overall cytochrome C levels do not change during apoptosis, only the localization which shifts from mitochondria to cytoplasm. Therefore, the results shown in Fig. 7A make no sense.

  1. Figure 7B shows that there is no apoptosis in the cells exposed to H2O2 (ie no increase in cells in lower right quadrant). Therefore, it is unclear why the authors talk about apoptosis throughout the manuscript.

  1. Since both Parkin and Pink 1 are mitochondrial proteins, their levels should be normalized to a mitochondrial housekeeping protein not actin since if mitochondrial levels decrease then the overall levels of Parkin and Pink 1 will also decrease but the levels per mitochondrion may not.

  1. line 468: There is no evidence that IBC increases NGF or BDNF secretion. The data show an increase within the cells, not in the medium.

  1. Figure 8: The authors say that IBC significantly increases the levels of BDNF, NGF and NT3 relative to H2O2 alone but that is only true for the highest concentration of IBC. The effects of the lower concentrations appear to be non-significant.

  1. The overall levels of Akt and mTOR are not particularly relevant. What is relevant is changes in their levels of phosphorylation which regulates their activity. Thus, either include phosphorylation studies or remove.

  1. Discussion: A number of statements do not agree with the data. For example, no evidence for mitochondrial ROS release is presented and as noted in #6, there is no evidence for apoptosis.

  1. Conclusion: The authors greatly overstate their conclusions. Why is this a "new fundamental mechanism of neuroprotection"? Many researchers have discussed maintenance of mitochondrial function as a mechanism of neuroprotection. Also, no evidence is presented that IBC restores mitochondrial function. This needs to be tested directly if the authors are going to claim this.

Author Response

Dear Reviewer

We were very pleased with the favorable reviews on our manuscript. We thank you for your thoughtful and helpful comments. We have extensively refined our presentation of the manuscript, figures, and references in accordance with your suggestions and the changes are marked in RED color in the revised manuscript. Below, please find our point-by-point responses to all raised queries.

Kind regards,

In-Hyuk Ha, M.D., Ph.D.

Round 2

Reviewer 1 Report

The quality of this manuscript is now improved.

Thanks

Author Response

Dear Reviewer

We deeply appreciate the time and effort that the reviewers have dedicated to providing your valuable feedback on my manuscript. Thank you for reading my work and providing reviewer's detailed feedback. We hope the revised manuscript will meet the requirements of academic publishing in Antioxidants

Kind regards,

In-Hyuk Ha, M.D., Ph.D.

Reviewer 3 Report

See atttached file.

Author Response

Dear reviewer

Thank you for giving us the opportunity to improve and resubmit our manuscript “Neurotherapeutic effect of Inula britannica var. chinensis against H2O2-induced oxidative stress andmitochondrial dysfunction in cortical neurons”.
Please find enclosed the revised manuscript for further consideration.
The manuscript hasbeen revised according to the comments raised by the reviewer to the best of our ability.
Changes to the manuscript are underscored and highlighted in red. Please find a detailed reply to the reviewer comments attached with this revision.

Kind regards,
In-Hyuk Ha, M.D., Ph.D.

Reviewer 4 Report

The authors have not sufficiently addressed my concerns. I stand by my original recommendation which was to reject the manuscript.

Author Response

 We apologize for not providing sufficient explanation in response to the reviewer’s comments submitted initially. And, we deeply appreciate the time and effort that the reviewers have dedicated to providing your valuable feedback on my manuscript. Thank you for reading my work and providing reviewer's detailed feedback.

Kind regards,

In-Hyuk Ha, M.D., Ph.D.
